# Transformer Monitoring with Electromagnetic Energy Transmission and Wireless Sensing

**DOI:** 10.3390/s24051606

**Published:** 2024-03-01

**Authors:** Shuxing Xu, Yurui Shang, Zhenming Li, Yongling Lu, Mingyang Liu, Wei Liu, Zhen Wang, Wei Tang

**Affiliations:** 1Beijing Institute of Nanoenergy and Nanosystems, Chinese Academy of Sciences, Beijing 101400, China; xushuxing@binn.cas.cn (S.X.); shangyurui@binn.cas.cn (Y.S.); 2Energy Storage and Electrotechnics Department, China Electric Power Research Institute, Beijing 100192, China; liumingyang@epri.sgcc.com.cn (M.L.); liuwei3@epri.sgcc.com.cn (W.L.); 3Research Institute, State Grid Jiangsu Electric Power Co., Ltd., Nanjing 211100, China; 15105182955@163.com (Y.L.); wangzhenscut@163.com (Z.W.)

**Keywords:** transformer diagnosis, light gas alarm, gas relay monitoring, electromagnetic energy transfer

## Abstract

To ensure stable and normal transformer operation, light gas protection of the transformer Buchholz relay is essential. However, false alarms related to light gas protection are common, and troubleshooting them often requires on-site gas sampling by personnel. During this time, the transformer’s operating state may rapidly deteriorate, posing a safety threat to field staff. To tackle these challenges, this work presents the near-field, thin-sliced transformer monitoring system that uses Electromagnetic Energy Transmission and Wireless Sensing Device (ETWSD). The system leverages external wireless energy input to power gas monitoring sensors. Simultaneously, it employs Near-Field Communication to obtain real-time concentrations of light gases, along with the electrified state and temperature. In field testing conducted on transformer relays’ gas collection chambers, the ETWSD effortlessly monitors parameters within warning ranges, encompassing methane gas concentrations around 1000 ppm, leakage voltage ranging from 0–100 V, and relay working temperatures up to 90 °C. Additionally, to facilitate real-time diagnosis for electrical workers, we have developed an Android-based APP software that displays current light gas concentrations, leakage voltage collection values, and temperature, while also enabling threshold judgment, alarms, and data storage. The developed ETWSD is expected to aid on-site personnel in promptly and accurately evaluating transformer light gas protection error alarm faults. It provides a method for simultaneous, contactless, and rapid monitoring of multiple indicators.

## 1. Introduction

In power systems, transformers are one of the most crucial components [1,2]. Any form of failure can lead to power supply disruptions, costly repairs, and even severe damage [3,4]. Transformers, after decades of continuous operation and exposure to varying operating conditions, face significant risks throughout their lifespan [4,5]. Therefore, protecting transformers is of utmost importance. Failures in oil-immersed power transformers are mainly categorized into external tank failures and internal tank failures [5,6]. Currently, electrical and non-electrical protections are used to isolate transformer faults to ensure their safe operation [2,7]. Electrical protection primarily involves forming a differential circuit through the currents on each side of the transformer [8], with protection schemes using current differences in the differential circuit under internal faults, external faults, and normal operation as criteria for action, aimed at protecting against various inter-phase short circuit faults in transformer windings and lead wires [1,7]. Non-electrical protection mainly refers to gas protection based on the large amount of free gas produced and the resulting oil flow surges during internal faults in the transformer tank. Light gas protection pertains to the degree of gas accumulation in the tank, while heavy gas protection relates to the speed of oil flow in the tank [9,10].

However, the accuracy of light gas action protection has been low over the long term [11,12]. This is because its criterion is the volume of gas accumulation, which is easily influenced by non-fault factors such as trapped gas, leading to insufficient reliability of protection [13,14]. As a result, power departments in many countries, including Europe and North America, typically use light gas relays as a means of alarm. The Chinese state grid has also established relevant regulations to improve the accuracy and reliability of light gas alarms. After light gas action, further confirmation is required, with manual on-site gas analysis [15,16]. Nevertheless, the transformer is still operational at this time, and in the event of a serious accident, it poses a significant safety hazard to workers. In response to this safety hazard, some scholars have conducted research on dissolved gases in oil and developed online monitoring technology for dissolved gases in oil. This technology, by analyzing changes in the composition of dissolved gases in oil under different voltages, temperatures, and other conditions, can accurately reflect early internal defects in transformers [17,18,19]. However, this method has a long monitoring cycle, usually measured in years, months, or weeks. It is not sensitive enough to respond promptly to the rapid development of internal defects into serious faults, leading to severe accidents. In summary, solving how to achieve rapid on-site qualitative diagnosis and avoid personnel casualties when staff perform gas extraction operations following minor gas alarms from transformers is particularly crucial. The current online diagnostic technologies mainly include gas chromatography and photoacoustic spectroscopy [20,21,22,23]. Nevertheless, both methods have long gas component detection cycles, typically in hours, and cannot sensitively and quickly respond to the rapid development of internal defects into serious faults. Additionally, both methods require the addition of extra equipment to the existing transformers, and the equipment is costly and complex to install.

This study analyzes the principles and prevailing issues in current light gas protection, presenting a novel Energy Transmission and Wireless Sensing Device System (ETWSD) grounded in Near-Field Communication (NFC) and wireless electromagnetic energy transfer. This method comprises an NFC part, a wireless electromagnetic energy transmission part, and a sensor part. It enables swift qualitative analysis of the methane gas concentration within the gas relay, the electrified condition of the gas relay’s casing, and the operating temperature of the gas relay under operational status. Energy is transferred through electromagnetic induction to charge a supercapacitor, which then powers the methane gas sensor for real-time monitoring of methane concentration in the transformer, achieving millisecond-level rapid response. NFC technology is employed for contactless collection of signals from the sensor unit, subsequently displaying real-time measurements of temperature, voltage, and methane gas concentration in the supervisory control APP. This ETWSD assists field personnel in assessing the current operational state of the transformer, thereby preventing rapidly developing faults during transformer operation and ensuring the safety of on-site workers.

## 2. Materials and Methods

### 2.1. Fabrication of Wireless Electromagnetic Power Transfer

The Wireless Electromagnetic Power Transfer (WEPT) design employs the SCT63240 chip from SCT Company (Franklin Lakes, NJ, USA), working in conjunction with a C51 microcontroller, to realize a wireless high-power transmitter system that conforms to the WPC (Wireless Power Consortium) specifications. This device integrates a 4-MOSFETs full bridge power stage, gate drivers, a 5 V step-down DC/DC converter, a 3.3 V Low Dropout Regulator (LDO), a 2.5 V accurate voltage reference, and an input current sensor for system efficiency. Two Pulse Width Modulation (PWM) signal input ports of this system can be controlled by the C51 microcontroller to operate the full bridge inverter across a wide frequency range of 20 kHz to 400 kHz, fully meeting the WPC specification frequency requirement of 100 kHz to 250 kHz. The typical application circuit for the SCT63240 chip can be obtained from its data manual. A corresponding Printed Circuit Board (PCB) is designed according to the Layout Guideline, and then the PWM1 and PWM2 pins of the chip are connected to the General-Purpose Input/Output (GPIO) ports of the C51 microcontroller. Within the C51 microcontroller, the timer is initialized, the PWM frequency is set, and PWM waveforms are generated by toggling the I/O port level states. The duty cycle of the PWM waveforms is adjusted by altering the duration of the high and low voltage levels, thus achieving control over the wireless charging frequency. Figure 1a shows the PCB layout of WEPT. The power supply routing uses a large-area copper laying method to reduce line impedance, resulting in better current-carrying capacity and higher energy transfer efficiency for the WEPT module. Figure 1b displays an optical image of the WEPT module, which comprises two components: the circuit board and the copper wire coil. Figure 1c shows the experimental test structure of the WEPT. A 3D-printed housing with a silicon steel sheet is fixed between the circuit board and the coil to block electromagnetic interference and suppress the generation of eddy currents. This stacking of the circuit board and coil increases space utilization.

### 2.2. Fabrication of NFC, Wireless Electromagnetic Power Receiver and Sensor Circuit

The Wireless Electromagnetic Power Receiver (WEPR) part of the circuit uses the CP2101 chip from COPO Company (Nacka kommun, Sweden), which complies with WPC specifications. The chip’s typical topology schematic and Layout Guideline are displayed in the chip’s data manual, and the circuit is designed according to the requirements of the chip manual. Figure 2a displays the circuit topology diagrams of the WEPR and methane sensor. The inductive coil antenna, controlled by the CP2101 chip, collects electromagnetic energy from the supervisory control. The energy is then rectified, filtered, and stored in a supercapacitor. The voltage of the supercapacitor is affected by the output current and voltage of the electromagnetic energy transmission system and is sensitive to environmental disturbances. To prevent overheating and damage to the methane gas sensor, a stable voltage supply is required. Therefore, a low-dropout linear regulator is added before the methane gas sensor to provide a stable voltage. Before normal operation, the methane gas sensor requires a 3 min preheating period, during which it needs a supply voltage of 2.4 V and a current of 32 mA. The WEPR system can quickly charge to meet the energy requirements for the sensor’s preheating, sensing, and communication.

Figure 2b shows that the NFC chip employs a minimal system peripheral circuit design using the RF430FRL152HCRGER (Texas Instruments, Dallas, TX, USA), with the NFC antenna designed using PCB printing technology. Impedance analysis is conducted using a vector network analyzer to match the impedance frequency values specified in the chip’s design manual. Additionally, a coupling capacitor is paralleled on the antenna, allowing the NFC antenna to reach the resonance frequency of the LC oscillation circuit at 13.56 MHz, while also serving to suppress interference. The temperature sensing module uses a thermistor-based measurement method. By measuring the voltage across the thermistor with the analog-to-digital converter (ADC) measurement port on the NFC chip, the resistance value changes with temperature, thus reflecting the actual environmental temperature. The voltage measurement module adopts a resistive voltage divider method for measuring high voltage. It uses resistors of 100 kΩ and 1 kΩ in series, and the NFC’s built-in ADC module collects the voltage value across the 1 kΩ resistor. The actual measurement voltage is then obtained through the resistive voltage divider formula. The methane gas measurement unit uses the MiCS-5524 MEMS sensor (DFRobot, Chengdu, China) with an analog output interface. The built-in ADC module of the NFC is used to collect and analyze the analog quantity.

### 2.3. Data Analysis and Statistics

All line graphs and curve graphs were created using Origin, while data processing was conducted in MATLAB R2023b. For distance testing between WEPR and NFC, temperature comparison tests, and long-term data display after ETWSD testing, the mean STD was used, with a sample size of 3. The error bars were too small to be displayed. After binary conversion and methane concentration calculation in MATLAB, the graphs were plotted in Origin.

### 2.4. Electrical Measurement and Characterization

The electrical signals of the ETWSD were measured utilizing a RIGOL MSO8204 oscilloscope (RIGOL, Suzhou, China). A RIGOL RP3500A probe was utilized for low-speed signal detection, while a RIGOL RP6150A probe was employed for high-speed signal analysis. Fixed frequency signals were generated using a STANFORD RESEARCH SYSTEMS MODEL DS345 instrument (SRS, Inc., Sunnyvale, CA, USA). Throughout the debugging and development phase of the ETWSD, a KEITHLEY 2200-72-1 programmable power supply (Tektronix China Ltd., Shanghai, China) was employed to provide stable voltage and current inputs. The output capacity of the ETWSD underwent meticulous calibration, facilitated by an A-BF DCT8730 electronic load meter (LCSC, Shenzhen, China). The recent redesign of the NFC signal antenna and the wireless charging coil necessitated recalibration of the antenna’s impedance characteristics, frequency response, and amplitude response. This recalibration was precisely executed utilizing a ROHDE&SCHWARZ ZNC3 model vector network analyzer (Rohde & Schwarz Hong Kong Ltd. (RSHK), Hong Kong, China).

## 3. Principle of Operation of the ETWSD

### 3.1. Light Gas Protection Alarm

When oil-immersed transformers are operational, light gas alarms can be triggered under several circumstances: Firstly, transformer oil gradually deteriorates over time due to factors like temperature and oxygen, leading to gas formation. Secondly, the insulation materials inside the transformer can age under high-temperature or -voltage stress, also resulting in gas production. Thirdly, partial discharges within the transformer, such as corona or spark discharges, can generate a small amount of gas. Fourthly, changes in oil level or severe fluctuations in oil temperature may also cause gas release. Fifthly, mechanical faults within the transformer, like loosening of windings, may trigger minor gas alarms. Sixthly, environmental changes around the transformer, such as temperature and humidity, or external gas infiltrating the system can occur. Seventhly, failure of insulation between windings can lead to turn-to-turn short circuits, phase-to-phase short circuits, or ground faults, producing a large amount of gas. Eighthly, the light gas relay itself might malfunction [24,25].

In case of insulation failure inside the oil tank of an oil-immersed transformer, the high-energy short-circuit arc generated at the fault point decomposes the transformer cooling oil and other insulation materials, thereby rapidly producing a large amount of free gas. The gas relay protects the transformer based on the volume of gas produced and the speed of oil flow it causes. Existing light gas protection is based on changes in the volume of free gas accumulated at the top of the gas relay, which is used to determine if there are minor faults inside the transformer tank. Typically, the action value of light gas protection is set according to the gas chamber volume within the gas relay, generally between 250 and 300 mL. A light gas alarm is triggered when the gas volume in the chamber exceeds this set volume.

After a light gas alarm is triggered, workers need to sample and test the gas while the transformer is in operation, a process that poses safety risks [26]. This is because the transformer’s operating condition may rapidly deteriorate during gas extraction, potentially leading to explosions and fires, posing a threat to the safety of the workers. In the current implementation process for false alarms related to light gas, gas extraction operations are permitted during transformer commissioning, provided that the operation process and steps are strictly followed. The non-contact auxiliary diagnosis carried out prior to this step also meets the safety distance requirements. This paper proposes a real-time qualitative diagnostic device powered by wireless electromagnetic energy, allowing workers to quickly assess the state of minor gas alarms in transformers.

After statistical analysis of the gas component data following light gas alarms in transformers as reported in the existing literature [24,27,28,29,30,31] (Table 1), it is found that under the correct operation of light gas protection, the proportions of hydrogen, methane, and ethylene gases in the gas components show noticeable changes. Among these, methane, as the gas with the most significant changes and highest distinctiveness, is used as the primary detection characteristic gas.

### 3.2. Component of ETWSD System

As shown in Figure 3a, this work presents a conceptual diagram of a device system based on Electromagnetic Energy Transmission and NFC wireless sensing, referred to as the ETWSD. The system comprises three parts: a smartphone installed with a development application, a wireless charging transmitter, and an ETWSD. The smartphone serves as the upper computer for the entire system. It obtains sensor data from ETWSD through NFC communication technology and displays the data graphically in the app using post-processing algorithms and display interface frameworks. The WEPT module is designed to transfer electrical energy from the phone to the ETWSD and can be omitted if the phone itself supports reverse wireless charging. The ETWSD module is an embedded system that integrates the functions of receiving electromagnetic energy, collecting physical quantities of gas relay operating status, and NFC communication.

Figure 3b shows the block diagram of the program structure of the ETWSD system. Both smartphones and computers can be used as the upper computer of the system, but the portability of smartphones gives them more advantages. The upper computer supplies energy to the WEPT module through a universal serial bus cable. The Wireless Electromagnetic Energy Receiver module enables the use of high-power sensors on board, and the sensor module collects the temperature, the electrified state of the casing, and the methane gas concentration in the gas relay. The NFC data module is responsible for data communication with the supervisory control APP. The WEPR module provides a stable power source for the sensor module. When the receiver is close to the transfer, the transfer and receiver communicate via metal coils, achieving a handshake function. The main controller continuously judges whether the circuit is operating in a normal energy transmission state, ensuring the efficiency and stability of energy transmission. At the same time, the main controller regulates the voltage adjustment module to filter, rectify, step-down, or step-up the input alternating electromagnetic signal through a series of voltage transformation operations. The main controller can set the output voltage and current of the voltage regulation module through external resistor programming. Under stable voltage drive, the gas sensor undergoes approximately 3 min of preheating. After preheating, it outputs an analog voltage signal to the NFC data module, enabling continuous concentration monitoring of specific gas components. The NFC data module collects analog data of temperature, voltage, and gas concentration through three analog data lines, converts them into digital form through an ADC, and then calculates the actual data values of temperature, voltage, and gas concentration through formulae. Subsequently, the RF modulation module modulates the data information into specific frequency electromagnetic signal information and transmits it wirelessly to the supervisory control. The supervisory control, after demodulating and recording the signal, finally displays it on the interface of the supervisory control application, thus completing the entire process from power supply to measurement and to display. Field workers can instantly view and export measurement data through the device.

Figure 3c shows the interface of the ETWSD supervisory control APP. The APP interface displays the connection status of the lower-level machine in real time through the Tag connection status, which can be used to check if the supervisory control is properly placed. The program monitors the amplitude of data from the three sensors and provides brief auxiliary reminders through the error alarm. Meanwhile, the interface displays real-time data line graphs of temperature, leakage voltage, methane concentration, and ADC port voltage. Measurement data can be saved using the SAVE button on the control panel. During transformer gas sampling operations, workers only need to place their smartphones in the sensing area and wait for the electromagnetic energy transmission to complete. After a few minutes, they can obtain the gas temperature inside the gas relay, voltage readings from leaks in the gas relay casing, and the concentration of methane in the gas relay collection chamber.

## 4. Experimental Results

### 4.1. Calibration of Methane Gas Sensors

Figure 4a displays the voltage and output current curves of the WEPR system under no-load conditions. The system maintains a stable output of 5 V, with the current gradually increasing to a maximum output value of 1.2 A within 60 s. The stable voltage and rapid current response enable the WEPR system to have good load-bearing capacity, providing the required energy expenditure for the sensor circuit.

Figure 4b displays the output voltage of the WEPR system at different distances from the electromagnetic energy transmitter. It can be observed that the WEPR system can stably receive wireless electromagnetic energy within 40 mm, as evidenced by the voltage output. Figure 4c illustrates the charging curves of the WEPR system’s internal supercapacitors with varying capacitance values. The time required to charge capacitors to 5 V varies depending on their capacitance. A 0.5 F capacitor takes only 15 s, a 1.5 F capacitor takes 41 s, a 5 F capacitor takes 67 s, and a 10 F capacitor takes 116 s. This rapid charging of capacitors can reduce the time needed for wireless electromagnetic energy transfer between the supervisory control and the lower-level machine. Figure 4d illustrates the correlation between the voltage on the supercapacitor in WEPR and the voltage on the methane sensor. The graph shows that the voltage on the methane sensor remains consistently stable at the expected value. A 10 F capacitor can provide power for the sensor’s preheating and working processes. Furthermore, it has been demonstrated that the system can provide sufficient power to the methane gas sensor for continuous measurement work lasting over 8 min, following a rapid charge of the storage capacitor. Figure 4e depicts the methane concentration in the ambient environment. Notably, at the 100 s juncture, upon opening of the gas supply valve, the sensor exhibits a prompt response within a 5 s timeframe, thereafter progressively aligning with the actual methane concentration observed in the environment. This expeditious response characteristic significantly enhances the measurement system’s overall speed and stability. Furthermore, Figure 4f presents an analysis of the correlation between the long-term operational data of the methane sensor and the corresponding calibrated values across varied methane gas concentrations. The empirical data display a marginal divergence from the calibration values, thereby underscoring the sensor’s precision and stability in measuring methane gas concentrations. It is, however, noteworthy that upon exceeding methane concentrations of 10,000 ppm, the sensor initially manifested a pronounced offset, which subsequently converged towards the actual calibrated values over time.

### 4.2. NFC Module Testing

To ensure the antenna has the appropriate shape and flexibility, we redesigned its shape and material. To achieve optimal signal and energy transmission, we used a vector network analyzer to establish the relationship between frequency and antenna impedance through frequency sweeping. By continuously adjusting the antenna’s design parameters, we finally obtained the antenna parameter curve shown in Figure 5a, matching the antenna’s maximum frequency and equivalent impedance with the chip’s port characteristics. In practical use, the distance between NFC antennas usually affects the accuracy of data transmission. As shown in Figure 5b, the data collected by the ADC completely match the actual data at distances less than 15 mm; beyond this limit, data transmission fails. In the sensor system, measurements of three types of data are carried out in analog form and converted into digital form through the ADCs. As shown in Figure 5c, the ADCs’ precision has been calibrated, providing valuable assistance in correcting the collected analog values. Within the 0–0.9 V range, the average error rate between the ETWSD’s voltage acquisition value and the power supply voltage is impressively low at 0.49%.

Figure 5d measures the comparison of the temperature sensor’s output values via the supervisory control with standard values, showing a notably lower average error rate, reaching as low as 0.36%, particularly at environmental temperatures below 70 °C. The measurement of leakage voltage with the casing electrified uses a resistive voltage divider method, where reference voltage data collected by the ADC can provide readings of the gas relay casing voltage. It can be seen from the graph that this method’s accuracy is acceptable, with a maximum error of less than 10%, as shown in Figure 5e. This is sufficient for the qualitative measurement of whether the gas relay casing is electrified. Figure 5f shows how we tested the relationship between the analog quantities collected by the ADC and methane concentration. Using professional methane concentration testing equipment, the curve of methane concentration varying over time was obtained. Additionally, an oscilloscope was used to measure the voltage change curve at the analog output port of the methane sensor. By fitting the curve with higher-order functions, the relationship between the test voltage analog quantity and the accurate methane concentration digital quantity was established, thereby creating a function mapping the voltage values to the analog quantities.

### 4.3. Laboratory Tests

The system’s current fault determination method utilizes the threshold comparison approach. If any of the three parameters exceed the designated threshold, the circular label on the APP will change from green to red, indicating abnormal monitoring sensing information. Figure 6a shows the actual working interface of the APP, which displays the connection status of the supervisory control. A green circle indicates there is no danger under the current data conditions, while a red circle signifies that at least one sensor’s current data are in an abnormal state. The interface also displays the current temperature, gas relay casing voltage, and methane gas concentration. The APP interface is clear and concise, facilitating quick access and judgment of key information.

Figure 6b depicts the actual working state in a laboratory setting. This sensing system operates in the collection chamber of the gas relay, using the glass panel of the collection chamber to complete electromagnetic energy transfer and NFC-based data communication. By controlling the temperature and methane gas concentration in a sealed beaker, the environment inside the actual gas relay collection chamber is simulated and reproduced as closely as possible, thereby testing the system’s stability and reliability. Figure 6c demonstrates the long-term stability and accuracy of the sensor system. However, the system mentioned above has not been able to operate in the gas relay of an actual high-voltage oil-immersed power transformer in operation. Although we have conducted experiments related to electromagnetic radiation interference, the magnitude of the interference noise is smaller than that of the electromagnetic interference noise emitted by the actual transformer. As a result, the quality of communication and electromagnetic transmission may be degraded in the actual working scenario.

## 5. Discussion

Under the current regulations and safety guidelines for gas relay protection operations, there still exist certain safety risks. Existing methods such as photoacoustic spectroscopy and gas chromatography are capable of effectively analyzing the components of dissolved gases in transformer oil, thereby providing early warnings for potential defects and issues in oil-immersed power transformers. However, the detection cycle of these methods is typically longer, usually exceeding 30 min. In cases of rapidly developing defects, they may not sensitively reflect the problems present in the transformer. Consequently, gas alarms may pose a threat to the personal safety of operators conducting gas sampling operations while the transformer is in operation. Furthermore, both existing methods require modifications to the transformer body and the installation of additional equipment, rendering them expensive and difficult to widely implement. The proposed method can be seamlessly integrated with existing gas relays. It enables rapid detection and analysis of methane gas concentration in gas relays, transformer temperature, and the electrification status of relays’ casings 3–5 min before personnel engage in gas sampling operations. This is aimed at ensuring the safety of on-site workers. Additionally, the proposed device features a compact size, low cost, and can be produced using flexible PCBs. Installing it in the gas collection chamber of gas relays would be an ideal scenario. Therefore, the proposed sensor device employs near-field communication and wireless power transmission technologies. The electromagnetic signals can penetrate the transparent window of the gas collection chamber, enabling interaction with handheld portable electronic devices.

The designed sensor circuit features a flexible antenna with impedance matching at 13.56 MHz, laying a stable foundation for wireless communication. Moreover, after preheating, the circuit can quickly respond to methane concentration, providing precise readings below 1000 ppm and qualitative detection above 1000 ppm. Additionally, it can swiftly detect the electrification status of gas relay casings, offering accurate readings in the range of 0–100 V. It also enables rapid measurement of transformer operating temperature within a temperature range of up to 90 degrees Celsius. However, the proposed sensor also has some drawbacks. For instance, it can only detect a limited range of gas types, which restricts its ability to provide detailed fault analysis in transformers, limiting its capability for qualitative detection. Furthermore, its ability to resist electromagnetic interference in communication remains unknown. Although we conducted relevant electromagnetic interference experiments, proving the reliability of near-field communication, we did not perform actual measurements in operational transformers. The electromagnetic environment in actual operating sites may be more complex and variable, which could affect the reliability of near-field communication.

## 6. Conclusions

In summary, this paper proposes a proximal, thin-slice transformer monitoring device using electromagnetic energy transmission and wireless sensing to assist workers in qualitatively diagnosing the operating condition of the transformer prior to gas extraction operations following the triggering of a light gas relay alarm. This device differs from existing online diagnostic technologies used in the power grid, such as gas chromatography and photoacoustic spectroscopy. While these previous methods require the addition of additional instrumentation or piping to the transformer, and require long-term testing, our method can detect minor defects and faults in the current operating condition of a transformer by analyzing dissolved gas components in the oil, helping to prevent transformer failures. In addition, these methods lack the agility to provide rapid diagnostic results in scenarios where internal defects develop rapidly or mechanical parts loosen, resulting in the ingress of external gases. In contrast, the proposed device can be placed in the gas transparent collection compartment of the gas relay and rapidly measures the operating temperature of the transformer, the leakage voltage of the gas relay housing and the methane concentration in the gas relay collection chamber. Together, these measurements assess the safety of the transformer’s current condition for on-site maintenance personnel performing gas sampling operations. The overall system design comprises three main parts: the supervisory controller, the NFC communication sensor data acquisition module and the wireless power supply module. The low-power sensors are powered by NFC, which provides communication functions while offering low power consumption. For the gas sensor, which requires a stable, continuous high voltage, electromagnetic wireless energy transfer and collection methods are used to charge the supercapacitor in a short time, ensuring normal operation of the gas sensor. Finally, by combining NFC communication technology with electromagnetic wireless energy transfer technology, we realized a device for rapid on-site auxiliary diagnosis of light gas relay alarms.

## Figures and Tables

**Figure 1 sensors-24-01606-f001:**
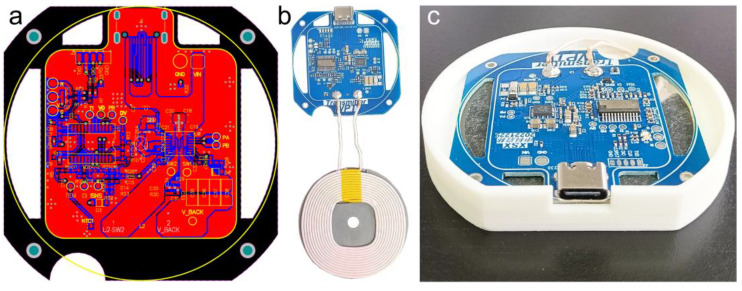
The schematic diagram of WEPT. (**a**) Altium designer layout diagram for WEPT. (**b**) Optical image of WEPT. (**c**) The diagram of WEPT for test structure.

**Figure 2 sensors-24-01606-f002:**
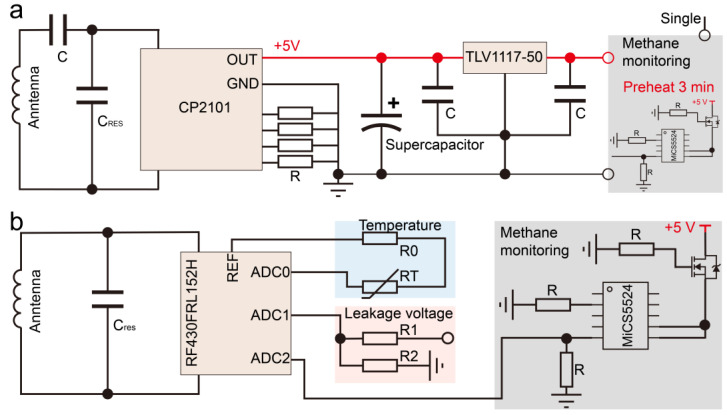
Partial circuit schematic of ETWSD. (**a**) Circuit topology diagram of WEPR. (**b**) Circuit topology diagram of the NFC circuit.

**Figure 3 sensors-24-01606-f003:**
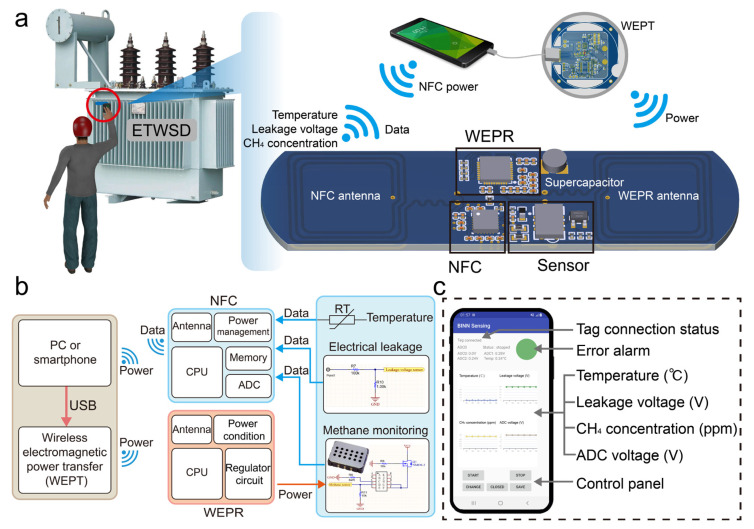
Conceptual and schematic diagrams of the ETWSD. (**a**) Conceptual diagram of the use of the ETWSD, along with schematic diagrams showing the basic structural components of smartphones and sensor nodes. (**b**) Block diagram of the signal and energy flow in the ETWSD system, highlighting the important components of each part. (**c**) Interface display of the supervisory control APP, including annotations for each display content.

**Figure 4 sensors-24-01606-f004:**
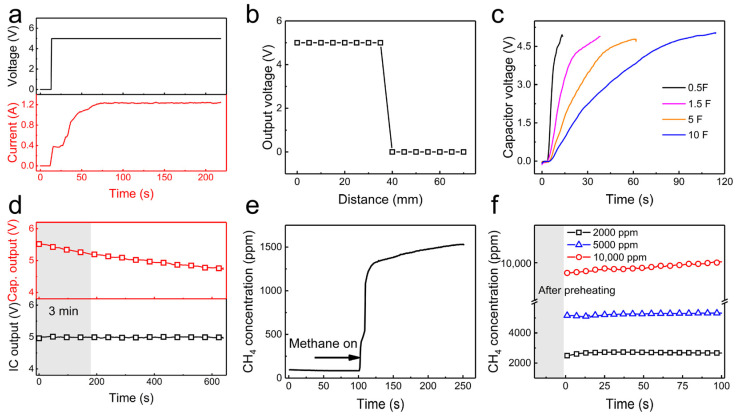
Wireless electromagnetic power receiver system. (**a**) Output voltage and current of WEPR under no-load conditions. (**b**) Relationship between WEPR output voltage and transmission distance. (**c**) Charging of different capacitance capacitors by the WEPR system. (**d**) Comparison of capacitor voltage and load voltage in WEPR under load conditions. (**e**) Methane concentration response test. (**f**) Actual test data of the methane gas sensor for different methane gas concentrations.

**Figure 5 sensors-24-01606-f005:**
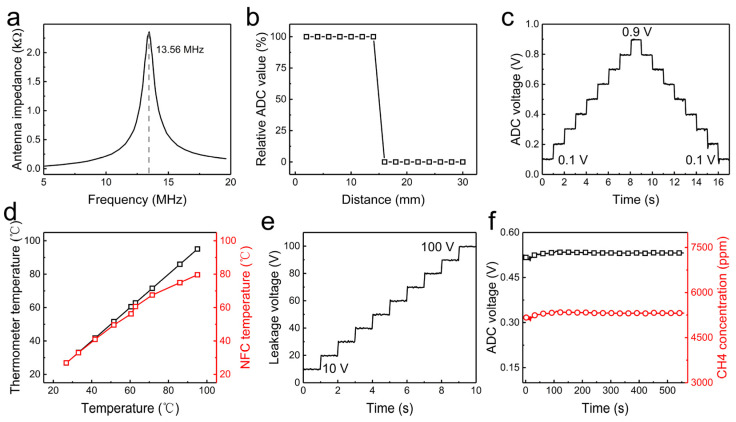
NFC data transmission module. (**a**) Antenna impedance frequency parameter diagram. (**b**) Correlation of data collected by the ADC. (**c**) Calibration of ADC collected voltage against actual values. (**d**) Comparison of temperature values collected by the temperature collection module with actual values. (**e**) Collection of leakage voltage values. (**f**) Calibration of ADC voltage values against methane concentration values.

**Figure 6 sensors-24-01606-f006:**
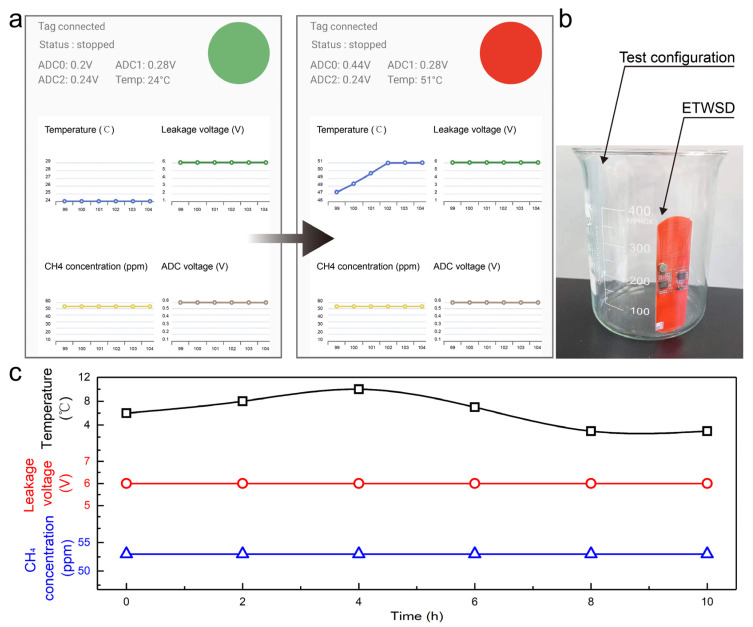
Laboratory test images of ETWSD. (**a**) Display pages of the APP in both normal and abnormal states. (**b**) Actual optical images of ETWSD under laboratory conditions. (**c**) Test data of sensor data from ETWSD under prolonged operation.

**Table 1 sensors-24-01606-t001:** Components of gas samples after light gas protection operation.

Protection Action Status	CH_4_ (ppm)	H_2_ (ppm)	C_2_H_4_ (ppm)	C_2_H_6_ (ppm)	C_2_H_2_ (ppm)
Error action	8.8	0.0	2.3	2.5	0.0
	30.7	257.0	5.6	2.5	0.2
	10.3	28.0	7.5	0.0	0.0
	5.9	928.3	1.4	0.1	14.2
Correct action	82.2	12,527.0	20.0	3.6	0.3
	2389.4	35,828.0	1225.4	0.0	3868.7
	3499.0	51,412.0	2302.0	7550.0	8121.0
	1484.0	70,391.0	83.0	2698.0	5171.0
	9380.3	109,728.0	9208.2	471.5	14,855.9

## Data Availability

The data presented in this study are available on request from the corresponding author.

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
