# Peer review of "Transformer Monitoring with Electromagnetic Energy Transmission and Wireless Sensing"

_sensors, 2024, doi:10.3390/s24051606_

Round 1

Reviewer 1 Report (Previous Reviewer 3)

Comments and Suggestions for Authors

The paper is well written. the authors have responded exhaustively to previous requests.

Author Response

Dear Reviewer:

We highly appreciate the reviewer’s positive comments on our work. And we also thank the reviewer’s detailed and responsible reviewing of our work.

Reviewer 2 Report (New Reviewer)

Comments and Suggestions for Authors

The current peer review work is interesting and contemporary in subject matter. It is clearly written and easy to read. But there is chaos in the work structure that needs to be corrected.

Section 2 should be corrected in its entirety. To it must be added certain passages and figures from Section 3:

1. A figure/diagram showing the Wireless Electromagnetic Power Transfer (WEPT) described in this subsection shall be added to subsection 2.1;

2. To subsection 2.2 figures 3a and 3b should be moved, together with the description to them. Their place is not in section 3 - results;

3. Ceate a new subsection 2.x, in which to move figure 2a together with the description to it from subsection 3.3;

4. Create a new section 3, for example titled "Principle of operation of the proposed sensor system" (authors can choose a more precise title, this is only indicative). Move figure 1 to this section, together with the descriptive part of subsections 3.1 and 3.2. Here, the authors will be able to explain in detail the principle of operation of their proposed new sensor system;

5. Section 3 to be renumbered to Section 4 and relevant subsections. The obtained experimental results should be presented there;

6. Create a new subsection 5 - Discussion. Here the authors analyze the obtained results;

7. Figure 4a is of very poor quality. Let the authors present it as a separate figure and emphasize in more detail the application they have created;

8. Some of the literature is from more than 5 years ago. Replace it with a newer one from the last 5 years.

Round 2

Reviewer 2 Report (New Reviewer)

Comments and Suggestions for Authors

What is written in section 5 is not appropriate for the discussion section. The text thus presented in this section is most appropriately moved to Section 4, as subsection 4.3, and perhaps titled "Laboratory tests" or some other more appropriate title as the authors see fit. In the "Discussion" section, a summary of the sensor presented by the authors should be given - advantages, disadvantages, summary of the results obtained, what is new in the sensor presented, etc.

What is written in this section should not be confused with the "Conclusions" section.

Author Response

This manuscript is a resubmission of an earlier submission. The following is a list of the peer review reports and author responses from that submission.

Round 1

Reviewer 1 Report

Comments and Suggestions for Authors

This paper introduces the EHWSD system, which employs electromagnetic energy harvesting to power the gas sensor, utilizes NFC wireless technology to sense gas concentration, temperature, and leakage voltage parameters, addressing the prevalent issue of light gas alarm faults in the power grids with transformers. Furthermore, a practical application scenario APP display system has been created, offering a strong assurance for real-time diagnosis and timely protection of transformers. The proposed monitoring of gas samples before the extraction process holds potential guiding value, especially in instances of light gas relay protection failure. Therefore, I recommend publishing this manuscript in Sensors, with a few minor corrections.

1.     In the “Materials and Methods” section, an omission is the detailed description of the experimental instruments employed. I suggest the author include this information to offer a comprehensive overview of the experimental setup.

2.     It is advisable to ensure consistency between the color of the curves in Figure 3d and f and the overall theme of the figure to prevent any confusion or doubts among readers.

3.     The manuscript currently exhibits several linguistic and grammatical issues. I recommend a comprehensive review and revision of the text to align it with the high standards of academic publishing.

Reviewer 2 Report

Comments and Suggestions for Authors

Diagnostics and monitoring of large power transformers is very important due to the negative consequences of a transformer emergency shutdown in the power system. In practice, a number of methods are used dedicated to the diagnosis of the electromagnetic circuit, insulation system, technical condition of the oil, mechanical integrity, etc.

The diagnostic and monitoring methods used are basically reliable and strict compliance of the transformer operators with periodic technical inspections does not cause any major problems.

This does not free scientists from searching for more reliable diagnostic methods.

However, I do not find any ideas for new research methods in this article. The article concerns rather a different method of measuring selected parameters that can be acquired from the Buchholz relay.

I have a series of the following general criticisms:

1.       The authors wrote that used energy harvesting technology. This would be true if they obtained this energy from the electromagnetic field around the transformer tank - the leakage electromagnetic field. The authors powered the sensor system wirelessly, but this is not energy harvesting.

2.       The distance over which the power supply energy is transmitted is approximately 40 mm, while the distance of transmitting measurement signals is 15 mm. Is this a safe distance? Wouldn't it be better to use battery-powered systems and transmit signals over a distance of several meters or even several dozen meters using the Bluetooth or Wi-Fi interface? Currently, some communication interfaces with low power consumption have been developed, which means that the battery lasts up to several months. When using photovoltaic panels, the battery can operate for up to several years.

3.       The Authors said in conclusions ( rows 337-338), that currently methods require the addition of extra measurement equipment. I disagree with the authors because the proposed solution also includes several additional devices/equipment. There is no information on how the measurement system will be incorporated into the Buchholz relay or other gas relay. Is this possible in existing relay designs?

4.       The Authors said in conclusions ( rows 341-342), that “…these methods lack the agility to swiftly provide diagnostic results…”. I disagree with this statement. My experience shows that proper monitoring and compliance with diagnostic procedures and operation within the scope indicated by the transformer manufacturer always ensures safe operation of the transformer.

5.       In extreme cases - short circuits close to the transformer terminals, overvoltages in the network due to lightning, large overloads - the transformer may be damaged - , but the system proposed by the authors will not protect it in these cases.

6. The paper shows that the measurement system was not tested in real transformer operating conditions. Therefore, the authors did not analyze the impact of electromagnetic interference from the transformer on the measurement system. Some time ago I performed measurements near a high-power transformer with a good quality recorder. It turned out that I only recorded noise - that's how big the electromagnetic field interference is. I suggest that the authors first check the electromagnetic compatibility issues of the proposed measurement system.I see the possibility of submitting modified version of this work again for publication in Sensors journal, but after taking into account the above comments.

The main remarks presented above are so critical that I will not quote further detailed remarks

Conclusion: As a reviewer, I cannot recommend this article for publication in this form.

Comments on the Quality of English Language

I had a number of difficulties understanding some paragraphs. It is advisable to use more correct and understandable technical statements.

Reviewer 3 Report

Comments and Suggestions for Authors

The paper is related to the developed EHWSD system for evaluating transformer faults and monitoring multiple indicators. The paper is well written. In my opinion, using NFC to monitor the state of the transformer in the power system has several issues. The NFC needs the operator to stay very near transformers, and this, in many cases, for safety issues can be very expensive because the operator cannot work without the suspension of the line. For example, direct Wi-Fi systems can be used with more distance between the transformer and the operator.

Moreover, wireless repeaters can extend the distance between the operator and transformer, and more than one transformer can be monitored simultaneously, collecting this information with a timestamp. In Europe, a similar system that uses a fiber optics or wireless connection called Stand Alone Merging Unit is used. The paper on this topic does not add a novelty. For this reason, the novelty of the paper should be improved.
